# Pyodermitis during Nivolumab Treatment for Non-Small Cell Lung Cancer: A Case Report and Review of the Literature

**DOI:** 10.3390/ijms24054580

**Published:** 2023-02-26

**Authors:** Terenzio Cosio, Filadelfo Coniglione, Valeria Flaminio, Roberta Gaziano, Deborah Coletta, Rosalba Petruccelli, Emi Dika, Luca Bianchi, Elena Campione

**Affiliations:** 1Department of Experimental Medicine, University of Rome “Tor Vergata”, 00133 Rome, Italy; 2Dermatologic Unit, Department of Systems Medicine, University of Rome Tor Vergata, Viale Oxford, 81, 00133 Rome, Italy; 3Department of Surgical Sciences, University Nostra Signora del Buon Consiglio, 1000 Tirana, Albania; 4Medical Oncology Unit, University of Rome Tor Vergata, Viale Oxford, 81, 00133 Rome, Italy; 5Laboratory of Microbiology, Microbiology and Virology Lab, Tor Vergata University Hospital, V. le Oxford, 81, 00133 Rome, Italy; 6Melanoma Center, Dermatology, IRCCS Azienda Ospedaliero-Universitaria di Bologna, 40126 Bologna, Italy; 7Unit of Dermatology, Department of Medical and Surgical Sciences, DIMEC, Alma Mater Studiorum, University of Bologna, 40126 Bologna, Italy

**Keywords:** immunotherapy, nivolumab, cutaneous infection, *Staphylococcus aureus*

## Abstract

Immunotherapy in oncology is replacing traditional therapies due to it specific action and limited side effects. Despite the high efficacy of immunotherapy, side effects such as bacterial infection have been reported. Bacterial skin and soft tissue infections represent one of the most important differential diagnoses in patients presenting with reddened and swollen skin and soft tissue. Among these infections, cellulitis (phlegmon) and abscesses are the most frequent. In most cases, these infections occur locally with possible contiguous spread, or as a multifocal manifestation, especially in immunocompromised patients. Herein, we report a case of pyodermitis in an immunocompromised district in a patient treated with nivolumab for non-small cell lung cancer. A 64-year-old, smoker male patient showed cutaneous lesions at a different evolution level in the left arm, all in a tattooed area, with one phlegmon and two ulcerated lesions. Microbiological cultures and gram staining revealed an infection caused by a methicillin-susceptible but erythromycin-resistant (ER-R), clindamycin-resistant (CL-R), and gentamicin-resistant (GE-R) *Staphylococcus aureus* strain. Despite immunotherapy becoming a milestone in oncologic treatment, more than the spectrum of immune-mediated toxicities of these agents needs to be investigated. This report highlights the importance of considering lifestyle and cutaneous background before starting immunotherapy for cancer treatment, with an emphasis on pharmacogenomics and the possibility of modified skin microbiota predisposing to cutaneous infections in patients treated with PD-1 inhibitors.

## 1. Introduction

Immunotherapy in oncology is replacing traditional therapies due to its specific action and limited side effects. The rapid and expanding development of monoclonal antibodies targeting checkpoint inhibitors in the Programmed death 1 (PD-1)/PD1 ligand (PD-L1) inhibitors have been introduced to treat non-small cell lung cancer (NSCLC) [1]. Because of the aggressive course of the disease, despite knowledge gained in tumor biology that has led to targeted therapies, the prognosis for patients with NSCLC is poor, with a five-year relative survival rate of 24% [2]. Nivolumab is licensed as second-line therapy for advanced NSCLC. At the same time, pembrolizumab, another PD-1 inhibitor, can be used as first-line treatment in the case of PD-L1 ≥ 50% as the cutoff to define PD-L1 expression positivity and as second-line when PD-L1 ≥ 1% of expression positivity [3,4]. In recent years, these therapies have led to a new adverse effect called immune-related adverse events (irAEs) [5]. Moreover, their usage and the alteration T-cell checkpoint blockade can favor the development of opportunistic infections, mainly in the upper respiratory tract, due to *Mycobacterium tuberculosis* and *Aspergillus fumigatus* [6]. Moreover, in the last five years, skin and soft tissue infections have also been reported in patients treated with nivolumab (Table 1) [7,8,9,10]. Among the cases in the literature, Tabchi et al. [10] reported a clear case of cutaneous infection due to *Staphylococcus aureus,* occurring as ecthyma gangrenosum of the right cervical region and torso. However, multifocal manifestation has not been described by other authors. Herein, we report a case of cutaneous staphylococcal infection in an immunocompromised district during nivolumab treatment for NSCLC.

## 2. Case Presentation

In this study, we present a case of NSCLC in a 64-year-old smoker (75-pack-year) patient with no other pathologies in anamnesis. In February 2015, he was visited for chronic dry cough in the Department of Respiratory Medicine at our hospital. Enhanced chest and abdominal computed tomography (CT) revealed a 2.3 cm mass in the superior lobe of right lung; thus, he underwent atypical lung resection. The anatomopathological analysis led to the diagnosis of NSCLC adenocarcinoma; Anaplastic lymphoma kinase (ALK) and epidermal growth factor receptor (EGFR) were negative. The tumor was classified as pT1N0M0 stage IAC according to the TNM classification of the Union for International Cancer Control (UICC) 8th edition [12]. After a one-year follow-up, he presented disease relapse in the site of the surgery with mediastinal and hilar lymph node metastasis. The Eastern Cooperative Oncology Group Performance Status (ECOG-PS) at the time of admission was 0. The patient received Pemetrexed and Cisplatin for six cycles. According to the Response Evaluation Criteria in Solid Tumors (RECIST), partial response was achieved. Furthermore, he experienced progressive disease (PD) in November 2017, when adrenal progression occurred. Thus, nivolumab was given as second-line treatment. Until today, after 54 therapy administrations, CT shows disease stability (Figure 1 and Appendix A). After 22 therapy cycles, the patient developed grade 2 hypophysitis and arthritis, assessed using the Common Terminology Criteria for Adverse Events version 5.0. Therefore, nivolumab administration was discontinued for eight weeks. Cycles of medication and treatment holiday were repeated, and the patient was carefully observed for irAEs. The patient returned to our attention in May 2022 for cutaneous lesions at different evolution levels in the left arm, all in a tattooed area, with one phlegmon and two ulcerated lesions (Figure 2). At clinical examination, we noticed an exudative lesion of the skin with a violet halo, an index of subcutaneous tissue infection compatible with suppurative bacterial infection. The patient said he had pain on the inspected site. The clinical diagnosis was phlegmon. Moreover, two other excavated lesions, arising in the tattooed area, with a clean margin and a fibrous bottom, were detected. There were also signs of inflammation of the subcutaneous tissues detectable through pain, tumor, and perilesional rubror. Circular areas of superficial flaking described by the patient as primary lesions followed by skin ulceration were detected. The microbiological examination revealed the presence of gram-positive cocci flora compatible with *Staphylococcus* spp. on the slide, organized in typical grape-like clusters. The cultures of pus from skin lesions confirmed the presence of *Staphylococcus aureus*. The antibiotic susceptibility assay, performed according to the European Committee on Antimicrobial Susceptibility Testing (EUCAST) criteria, revealed a strain of *Staphylococcus aureus* multidrug-resistant to clindamycin (0.25), erythromycin (>4), and gentamicin (>4) but susceptible to methicillin (<0.25) (Appendix A). At baseline, T0, other lesions, such as an erythematous plaque and a scaly redness papule were noticed but with no evolution. The patient was treated with topical fusidic acid combined with oral clarithromycin 250 mg/daily for 15 days, with complete resolution of the lesions and no symptoms at the follow-up visit (Figure 3).

## 3. Discussion

Lu et al. [13] in their review reported that current large randomized clinical trials had not shown any increased risk of infection for patients receiving PD-1/PD-L1 inhibitors. Pneumonia is the most common infection in patients with irAEs, followed by skin and soft tissue infections (SSTI) and urogenital tract infections (UTI) [12]. The most frequent patterns of cutaneous toxicity caused by PD-1 inhibitors are lichenoid dermatoses, pruritus, and vitiligo [14,15]. Other dermatoses such as bullous pemphigoid, psoriatic reactions, and subacute cutaneous lupus erythematosus have been reported [16,17]. Although these clinical manifestations are included in irAEs, local or systemic infections must be considered in this fragile population, especially in patients with a medical history of recurrent infections or anatomical alterations referred to as *Locus Minoris Resistentiae* (LMR). The concept of *Locus Minoris Resistentiae* is an old but still effective way of thinking in Medicine. In dermatology, there are many reports of privileged localization of cutaneous diseases on damaged skin, which therefore represents a typical condition of LMR. The innovative concept of the immunocompromised cutaneous district (ICD) has been recently introduced to explain why a previously injured cutaneous site may become a privileged location for the outbreak of opportunistic infections, tumors, and immune reactions [18]. It is well known that all types of cutaneous scars are vulnerable sites for the development of neoplasms, infections, and dysimmune reactions. The complex underlying mechanisms have been recently included in the concept of the ICD [19]. This term indicates a regional immune dysregulation caused by the failure of lymph flow or altered neuropeptide release [20]. An interesting example of Ruocco’s immunocompromised district has been reported by Verma [21], who described a superficial dermatophytosis (tinea) restricted to tattooed sites. Moreover, some interesting articles that observed molluscum contagiosum and verrucae preferentially appearing in black tattoos imply the possibility of black tattoo ink reducing cellular as well as humoral local immunity [22,23,24]. The manifestations of skin infection occurred more severely in the tattooed area, thus confirming the hypothesis that the local microtrauma in adjuvant with the pigment created a microenvironment favorable to the infection, therefore confirming the ICD theory. Carbon black tattoo could be deposited in the basal epithelial layer, similarly to melanin granules [25]. These particles tend to settle on the epithelial cells’ supranuclear areas miming melanin distribution. Balfour et al. [26] discussed the role of black ink in granulomatous tattoo reaction. Kazandjieva et al. [27] also supposed a photoallergic effect from the tattoo pigment. Moreover, Lehner et al. reported the malignancies correlated with tattoos suggesting that ultraviolet radiation, through photoallergic effect, could induce a persistent inflammatory reaction, or that even trauma may promote malignant transformation [28]. This suggestion is linked to the fact that many Polycyclic Aromatic Hydrocarbons (PAH) such as benz[a]an-thracene are not only mutagenic but also generate reactive oxygen species (ROS). When tattooed skin is exposed to solar radiation, the ultraviolet part of the spectrum can be absorbed in the present benz[a]anthracene, which generates singlet oxygen with a quantum yield of 85% [29,30]. Based on this evidence, and considering immunotherapy, other authors report the arising of adverse events related to immunotherapy in tattooed areas. Monibi et al. [31] reported a case of subcutaneous sarcoidal granulomatous inflammation in a 58-year-old male presented with lesions occurring only within his black tattooed skin on the chest, shoulders, back, left forearm, and right thigh for the past three months during ipilimumab treatment. Kim et al. [32] presented a case of Lofgren syndrome sarcoidosis, which was first seen on a tattoo in a patient with metastatic urothelial cancer on therapy with ipilimumab and nivolumab. Yao et al. [33] reported a case about a 51-year-old man who came to them with extremely pruritic dermal papules within the red-inked areas of a tattoo on his right shoulder made 30 years before and previously treated with ipilimumab and nivolumab, followed by single-agent nivolumab, for a metastatic renal cell. These cases reported in the literature underline how not only infections but also granulomatous reactions can arise on tattoos during immunotherapy treatment and highlight the tattooed areas as sites consisting of a peculiar milieu more subject to alterations of the physiological state. They also pave the way for future studies of loco-regional immunity and intercellular crosstalk, especially between immune cells and fibroblasts.

The tattoo, both in its implementation using skin microtraumas and for the deposit of pigment, in the photo-exposed areas, induces greater fibrotic and elastolytic reactions with a thickening of the dermis and a decrease in the superficial perfusion of the epidermis. This leads to a local alteration of both trophism and local immunity, thus favoring the development of bacterial infections, according to the ICD theory (Figure 4). Among the most common bacterial pathogens responsible for skin opportunistic infections, *Staphylococcus* spp. are the most important and abundant skin-colonizing bacteria of healthy individuals. *Staphylococcus* spp. can cause a wide spectrum of skin and soft tissue infections ranging from superficial infections such as impetigo, erysipelas, folliculitis as well as deeper infections such as abscesses and phlegmons. Microabrasions compromise the skin’s microbial barrier function and may predispose to Staphylococcal cutaneous infections. This case of skin infection caused by *Staphylococcus aureus*, developed within a large tattooed area, illustrates very well the principle of LMR representing the tattooed area as a privileged site for the onset of *Staphylococcus* opportunistic infection. However, we cannot rule out the possibility that, in this case, the treatment with immune checkpoint inhibitors, which play an important role in the development of infectious diseases by limiting harmful inflammatory responses, might have contributed to increased susceptibility to cutaneous infections. The PD-1 pathway plays a critical role in regulating self-tolerance. In murine models, blocking the PD-1 pathway via genetic knock-down or through the administration of blocking antibodies increases the risk for developing autoimmune dilated cardiomyopathy and experimental autoimmune encephalomyelitis [34,35]. These in vivo data show how the alteration of the PD-1 pathway can lead to an alteration of self-tolerance. More specifically, considering ICDs, but above the entire tattooed area, the state of immunotolerance towards the exogenous pigment is lost, exposing the site to a possible autoimmune reaction with dysregulation of the skin barrier and a possible infection from bacteria colonizing the skin. Moreover, in our case, the patient was a heavy smoker, a renowned exacerbating factor in *Staphylococcus* colonization. Cigarette smoke (CS) exposure induces staphylococcal biofilm formation in an oxidant-dependent manner. CS treatment induced transcription of *fnbA* (encoding fibronectin binding protein A), leading to an increased binding of CS-treated staphylococci to immobilized fibronectin and increased adherence to human cells [36].

## 4. Conclusions

This case illustrates the principle of LMR, as demonstrated by skin staphylococcal infections that were found in old tattoos, while normal skin remained uninvolved. Although microenvironment and genetic factors seem to be involved in the pathophysiology of LMR, in our case, immune checkpoint inhibitor therapy may also play an important role in favoring the onset of *Staphylococcus aureus* opportunistic infection in an immunocompromised cutaneous district. Although immunotherapy is now a milestone in oncologic treatment, the spectrum of immune-mediated toxicities from these agents needs to be investigated. This report highlights the importance of considering lifestyle and cutaneous alteration before taking into consideration immunotherapy when planning an appropriate cancer therapy.

## Figures and Tables

**Figure 1 ijms-24-04580-f001:**
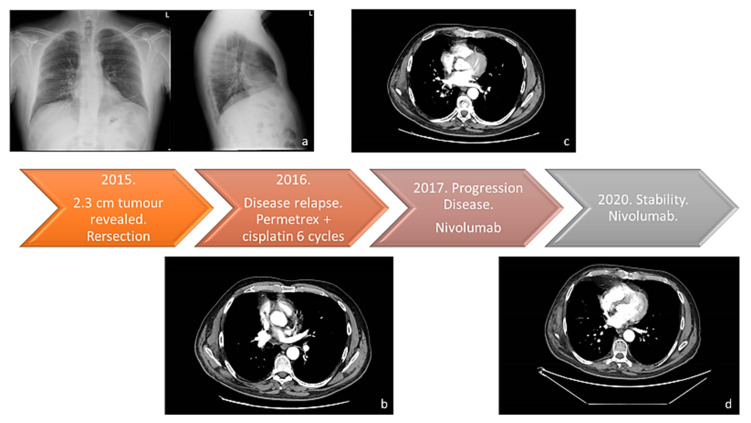
The therapy response timeline from 2015 to today. (**a**) Chest X-ray (Thoravision). Examination performed with digital technologies in the two orthogonal projections. The parenchymal opacity already indicated on the right and better evident in the previous TC of 3/3/2015 is confirmed. Disventilatory streak in the left basal site. (**b**) Examination performed before and after intravenous administration of organ-iodized contrast medium place comparison with the previous PET-CT investigation carried out on 29 July 2016. Compared to the previous one, there is a documented increase in the size of the newly formed tissue located in the right hilar seat (current dimensions 27 × 25 mm vs. 20 × 11 mm of the previous exam). The findings in the lungs are unchanged, the thickening area previously described at the level of the LID. No pleural effusion flaps. (**c**) Examination carried out before and after administration of organ-iodized MDC by IV route, compared with previous PET-CT carried out on 5 July 2018. There are no areas of pathological enhancement in the sub- and supratentorial sites. The axial ventricular system showed dimensions within the limits of the norm. The results of the surgery known in the anamnesis in the thoracic area on the right have been confirmed. The millimetric nodular formation already evident at the lower lobe appears unchanged right in the previous aforementioned signs of emphysema in both parenchymal areas. No pleural effusion flaps have been observed. (**d**) Neck-thorax: Nodular thyroid isthmic formation requiring monitoring remains unchanged instrumental with ultrasound examination. The results of surgery on the upper-right lobe have been confirmed. The share of tissue in the right hilar seat and the solid nodular formation (4 mm) correspond to the LID. No pleural or pericardial effusion. Sub-centimetric lymph node formations in the hilar and mediastinal region were stable in size.

**Figure 2 ijms-24-04580-f002:**
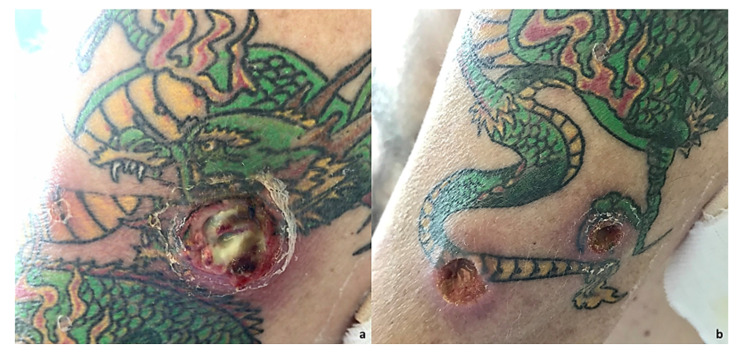
(**a**) At an objective examination, it is possible to notice an exudative lesion of the skin with a violet halo index of subcutaneous tissue infection compatible with a suppurative bacterial infection. The patient stated he had pain on the inspection site. Moreover, pain, *tumor*, *calor*, *rubror*, all signs of acute inflammation, were detectable. Clinical diagnosis of phlegmon was made. (**b**) On physical examination, it is possible to notice two lesions arising in the tattooed area. The lesions are excavated, with a clean margin and a fibrous bottom, where it is possible to see the onset of skin infection; then, infection is confirmed through microbiological tests. Clinical signs of subcutaneous tissue inflammation, i.e., pain, tumor, and perilesional *rubror* were detectable. Circular areas of superficial flaking, as reported by the patient, also occurred as primary lesions followed by skin ulceration.

**Figure 3 ijms-24-04580-f003:**
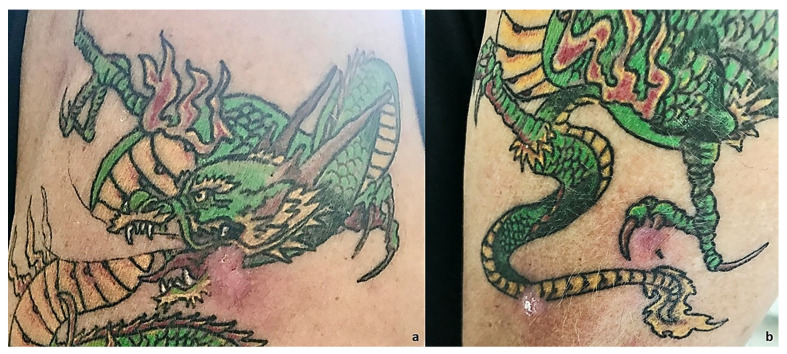
(**a**) Clinical resolution of the phlegmon after 15 days. (**b**) Clinical resolution with topical fusidic acid in combination with oral clarithromycin 250 mg/daily for 15 days.

**Figure 4 ijms-24-04580-f004:**
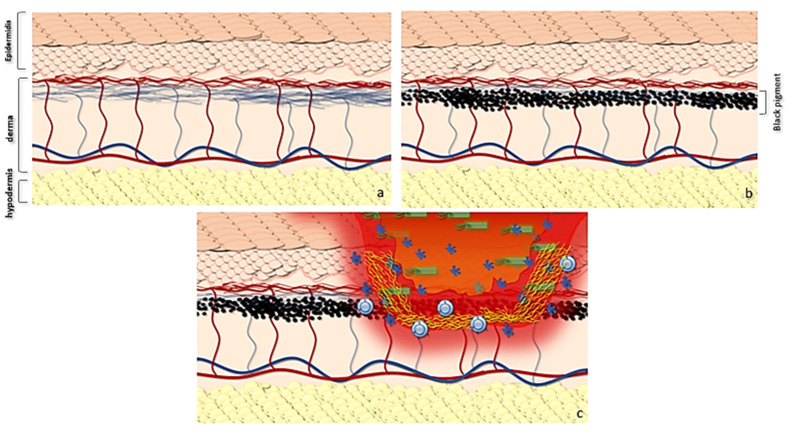
Graphic abstract showing the possible evolution of gram+ skin infections in tattooed patients with immunotherapy. (**a**) Normal skin anatomy and trophism of epidermis. (**b**) The use of the pigment leads both to a local microtrauma and to the deposit of pigment in the superficial portion of the dermis with processes of inflammation at the time of tattooing and subsequent inflammation. At the same time, the pigment, together with microtrauma fibrosis, reduces the diffusion of transudate from the dermis to the epidermis, also decreasing the superficial trophism. (**c**) This determines a reduced barrier action of the epidermis, which supports Ruocco’s hypothesis. In LMR, infections can lead to a local destructive process evolving in phlegmon.

**Table 1 ijms-24-04580-t001:** The table shows the cases of skin infections described in the literature according to the search criteria reported in the Appendix A search strategy section.

Reference	Clinical Condition	Clinical Presentation	Clinical Diagnosis	Number of Patients (Percentage)	Body Site	Drugs	Microbiological Identification	Antibiotic Resistance
Bavaro et al. [7]	Stage IV lung cancer	ND	Skin and soft tissue infection	2/12 (17%)	ND	Pembrolizumab	methicillin-sensitive *Staphylococcus aureus*	ND
Karam et al. [8]	ND	erysipelas	Skin and soft tissue infection	7/36 (19.4%)	ND	Anti PD-1/PD-L1 not specified	ND	ND
Fujita et al. [9]	Stage IV lung cancer (non-small cell)	ND	Skin and soft tissue infection	2/166 (1.2%)	ND	Nivolumab	methicillin-sensitive *Staphylococcus aureus*	ND
Ross et al. [10]	ND	ND	Skin and soft tissue infection	8/112 (7.2%)	ND	Nivolumab Pembrolizumab Ipilimumab	1 *Actinomyces radingae* 1 *Enterobacter cloacae* 1 *Enterococcus faecalis* 2 methicillin-resistant *Staphylococcus aureus* 3 ND bacteria	Methicillin
Tabchi et al. [11]	Stage IV lung cancer (adenocarcinoma)	ecthyma gangrenosum	Skin and soft tissue infection	1/1 (100%)	right cervical region and torso	Nivolumab	*S. aureus*	ND

## Data Availability

All relevant information is presented in the case report. Any additional data may be made available on reasonable request from the corresponding author.

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
