# Peer review of "Pyodermitis during Nivolumab Treatment for Non-Small Cell Lung Cancer: A Case Report and Review of the Literature"

_ijms, 2023, doi:10.3390/ijms24054580_

Round 1

Reviewer 1 Report

The case report was well written. Could authors also check if the previous reported skin infection in patient received immunotherapy also happened in the tattooed area?

Author Response

Dear Reviewer,

Thank you for the positive feedback on our case, hoping to make the consultation of skin infections from immunotherapies easier in clinical practice and in teamwork for cancer patients. With reference to your question, from the data reported in literature, there is no mention of tattooed patients in clinical trials, while there are no case report describing the onset of infections on tattoos. We evaluated each article with the relevant clinical photos in which there were tattoos. Following your request, we searched on PubMed and Scholar to see if there was a correlation between immunotherapy and other tattoo reactions. After our literature search, we identified three articles, which we also reported in the discussion of the case report, concerning the onset of adverse reactions (non-infectious) in skin areas tattooed during immunotherapy. The cases reported in literature underline how not only infections, but also granulomatous reactions, can arise on tattoos during immunotherapy treatment and highlighted the tattooed areas as sites consisting of a peculiar milieu more subject to alterations of the physiological state. This paves the way for future studies of loco-regional immunity and intercellular crosstalk.

“Monibi et al. [31] reported a case of subcutaneous sarcoidal granulomatous Inflammation in a 58-year-old male presented with lesions occurring only within his black tattooed skin on the chest, shoulders, back, left forearm, and right thigh for the past 3 months during ipilimumab treatment.  Kim et al. [32] presented a case of Lofgren syndrome sarcoidosis, which was first seen on a tattoo in a patient with metastatic urothelial cancer on therapy with ipilimumab and nivolumab. Yao et al. [33] reported a case about a 51-year-old man came with extremely pruritic dermal papules within the red-inked areas of a tattoo on his right shoulder made thirty years before and previously treated with ipilimumab and nivolumab, followed by single-agent nivolumab, for a metastatic renal cell. These cases reported in the literature underline how not only infections, but also granulomatous reactions, can arise on tattoos during immunotherapy treatment and highlight the tattooed areas as sites consisting of a peculiar milieu more subject to alterations of the physiological state. They also pave the way for future studies of loco-regional immunity and intercellular crosstalk, especially between immune cells and fibroblast”.

Reviewer 2 Report

In this manuscript entitled “Pyodermitis during nivolumab treatment for non-small cell lung cancer: a case report and review of the literature”, Cosio, et al. reported cutaneous staphylococcal infections during nivolumab treatment for non-small cell carcinoma and reviewed literatures relating to immune-related adverse reactions. Although an increased risk of opportunistic infections has not been statistically confirmed during the treatment with PD-1 immune checkpoint inhibitors, some reports demonstrated the incidence of infections such as upper tract infections caused by Mycobacterium tuberculosis and Aspergillus fumigatus during the PD-1 immune checkpoint therapy. Recent reports have further shown that treatment with PD-1 immune checkpoint inhibitors caused skin and soft tissue infections. The authors’ finding is, therefore, very interesting and informative. One important thing is that the authors should present possible molecular mechanisms underlying the onset of staphylococcal infections during the treatment with PD-1 immune checkpoint inhibitors by focusing on the immunocompromised cutaneous district model.

Minor comments:

Page 8, Line 179: “, too” should be removed.

Author Response

Dear Reviewer,

Thank you for the positive feedback on our case, hoping to make the consultation of skin infections from immunotherapies easier in clinical practice and in teamwork for cancer patients. With reference to your question, we described a possible immunological pathway that can contribute to infections based on PD-1 signaling in the light of the patient’s habits.

“The PD-1 pathway plays an critical role in regulating self-tolerance. In murine models, blocking the PD-1 pathway via genetic knock-down or through the administration of blocking antibodies increases the risk for developing autoimmune dilated cardiomyopathy and experimental autoimmune encephalomyelitis [34,35]. These in vivo data show how the alteration of the PD-1 pathway can lead to an alteration of the self-tolerance. More specifically, considering ICDs but above the entire tattooed area, the state of immunotolerance towards the exogenous pigment is lost, exposing the site to a possible autoimmune reaction with dysregulation of the skin barrier and a possible infection from bacteria colonizing the skin. Moreover, in our case, the patient was a heavy smoker, a renowned exacerbating factor in Staphylococcus colonization. Cigarette smoke (CS) exposure induces staphylococcal biofilm formation in an oxidant-dependent manner. CS treatment induced transcription of fnbA (encoding fibronectin binding protein A), leading to an increased binding of CS-treated staphylococci to immobilized fibronectin and increased adherence to human cells [36].”

Minor comments:

Page 8, Line 179: “, too” should be removed.

Dear Reviewer,

Thank you for the suggestion. The manuscript has been re-edited by a native English and line 179 corrected.

Best Regards,

Cosio T. and co-authors
